

# Side-effects of hyperthermic intraperitoneal chemotherapy in patients with gastrointestinal cancers

Jiyun Hu[1], Zhenxing Wang[2], Xinrun Wang[1] and Shucai Xie[1,2]

[1] Department of Critical Care Medicine, National Clinical Research Center for Geriatric Disorders, Xiangya Hospital, Central South University, Changsha, Hunan, China
[2] Department of Hepatobiliary Surgery, Haikou People's Hospital/Affiliated Haikou Hospital of Xiangya Medical College, Central South University, Haikou, Hainan, China

## ABSTRACT

**Background:** Hyperthermic intraperitoneal chemotherapy (HIPEC) produces unwanted side-effects that are mainly caused by chemotherapeutic drugs in the treatment of gastrointestinal (GI) cancers, and these effects have not been systematically summarized. The aim of this article was to provide a comprehensive overview of the side-effects of HIPEC for GI cancers and propose practical strategies for adverse event management.

**Methodology:** PubMed, Web of Science, and the Cochrane Library were systematically searched for side-effects of HIPEC in GI cancers prior to October 20, 2022. A total of 79 articles were included in this review.

**Results:** Adverse events, such as enterocutaneous digestive fistulas, GI tract perforation, neutropenia, postoperative bleeding, ventricular tachycardia, hyperglycemia, hypocalcemia, renal impairment, encapsulating peritoneal sclerosis, scrotal ulceration, and sarcopenia were described, and their clinical management was discussed. These side-effects involve the digestive, hematopoietic, circulatory, metabolic, and urinary systems. Effective methods for adverse event management included an expert multidisciplinary team, replacing chemotherapy drugs, using Chinese medicine, and careful preoperative assessments.

**Conclusion:** The side-effects of HIPEC are frequent and can be minimized by several effective methods. This study proposes practical strategies for adverse event management of HIPEC to assist physicians in choosing the optimal treatment method.

## INTRODUCTION

Cancer is a major public health problem and the leading cause of death worldwide, with approximately 19.3 million new cases and 10.0 million deaths globally in 2020 (*Sung et al., 2021*). Gastrointestinal (GI) cancers account for 26% of the global cancer incidence and 35% of all cancer-related deaths (*Arnold et al., 2020*). The pathophysiology and pathogenesis of GI cancer is complex and multifactorial. Microbiota dysbiosis, unchecked inflammasome activities perpetuating chronic inflammation, the cyclooxygenase

Corresponding author
Shucai Xie, 282791444@qq.com

(COX)–2/prostaglandin (PGE)2 pathway, and excess adiposity play important roles in the molecular and pathophysiological basis of GI cancers (*LaCourse, Johnston & Bullman, 2021*; *Man, 2018*; *Murphy, Jenab & Gunter, 2018*; *Sender, Fuchs & Milo, 2016*). With the development of medical technology and a deeper understanding of the pathogenesis of GI cancers, current therapeutic modalities for the treatment of GI cancers include surgery, chemotherapy, radiotherapy, and immunotherapy. Hyperthermic intraperitoneal chemotherapy (HIPEC), an emerging therapeutic modality, is currently used as an essential component of treatment, to improve the disease-free and overall survival of patients with primary and metastatic GI cancers (*González-Moreno, 2006*; *Klempner & Ryan, 2021*; *Loggie & Thomas, 2015*; *Verwaal et al., 2008*).

In 1980, *Spratt et al. (1980)* first performed HIPEC-based treatment in a patient with pseudomyxoma peritonei; thereafter, the use of HIPEC was explored in patients with GI cancers (*Hirose et al., 1999*; *Kaibara et al., 1989*). In 1994, *Hamazoe, Maeta & Kaibara (1994)* used mitomycin C (MMC) at 10 mg/mL to prevent peritoneal recurrence of gastric cancers. As soon as the abdomen was closed after gastric resection, patients were administered this treatment while under general anesthesia on the operating table. There is a 53–66% probability that a patient with metastatic gastric cancer will develop peritoneal metastases (PM) (*Dong et al., 2019*). HIPEC along with cytoreductive surgery (CRS) is the only therapeutic modality that has resulted in long-term survival in specific groups of patients. As a palliative treatment in advanced PM with intractable ascites, HIPEC has been shown to control ascites and reduce the need for frequent paracentesis (*Goéré et al., 2013*; *Seshadri & Glehen, 2016*). HIPEC plus CRS achieved great survival benefits in patients with peritoneal cancer (PC) of colorectal origin (12.6–22.3 months). However, this treatment also produced several side-effects which greatly hinder the application of HIPEC in the treatment of GI cancer (*Mancebo-González et al., 2012*; *Stiles et al., 2020*; *Verwaal et al., 2003*).

The side-effects of HIPEC (mainly caused by chemotherapeutic drugs) in patients with GI cancers are poorly defined. This review describes the current knowledge regarding the mechanism of action, safety, and side-effects of HIPEC in the treatment of GI cancers and explores the current knowledge gaps. With the aim of improving preoperative planning, preventing morbidity, and enhancing surveillance, we provide physicians with the latest information to assist them in choosing the optimal method of combined or primary treatment and predict which patients are at risk of experiencing side-effects.

## SURVEY METHODOLOGY

We systematically searched for relevant studies in PubMed (1,050), Web of Science (1,628), and the Cochrane Library (203) prior to October 20, 2022. The Medical Subject Headings or key words used were the following: ("hyperthermic intraperitoneal chemotherapy," or "intraperitoneal thermo-chemotherapy," or "HIPEC,") AND ("gastrointestinal cancers," or "GI cancers," or "gastrointestinal tumors" or "GI tumors," or "Gastric Cancer," or "Colon Cancer," or "Rectal Cancer," or "Colorectal Cancer," or "Appendiceal Cancer," or "Peritoneal Cancer"). The exclusion criteria included the following: duplicate literature, and literature not specified in these key words, comprising

other types of cancer patients, other cancer indices, other study outcomes, and studies lacking original or complete data. Table 1 shows the 16 studies selected which cover the typical side-effects of GI cancers.

## Rationale for hyperthermic intraperitoneal chemotherapy

The rationale of HIPEC is based on the concept of peritoneal dialysis; when chemotherapy is retained in the peritoneal cavity by the peritoneal-plasma barrier, small nodules of cancer are exposed on the abdominal and pelvic surfaces (*Dedrick et al., 1978*; *Flessner, 2005*). Normal tissue cells can withstand 47 °C for 1 h under high-temperature conditions, while malignant tumor cells can only withstand 43 °C for 1 h (*Garofalo et al., 2006*). Large-volume perfusate-containing chemotherapeutic drugs are heated to a certain temperature and continuously circulate, then remain for a certain period of time in the abdominal cavity of the patient, which can effectively kill and remove the residual cancer cells and minute lesions in the body cavity (*Kusamura et al., 2008*; *Ye et al., 2020*). Various methods for delivering HIPEC have been proposed, all of which are variations of two modalities: the open and closed techniques. The open technique ensures optimal distribution of heat and cytotoxic solutions, with the disadvantages of heat loss and leakage of cytotoxic drugs. The closed technique prevents heat loss and drug spillage and increases drug penetration but does not ensure homogeneous distribution of the perfusion fluid (*Lotti et al., 2016*).

The theoretical basis of HIPEC for GI cancers is as follows: first, tissue penetration of the intraperitoneal chemotherapy is facilitated by moderate hyperthermia (41–42 °C) (*Sugarbaker, Van der Speeten & Stuart, 2010*). When the temperature at a tumor site is >42 °C, cell killing phenomena are evident, such as destruction of the cell membrane, denaturation of proteins, and irreversible damage of tumor cells, while normal cells remain intact (*Sticca & Dach, 2003*; *Zhang et al., 2016*). Second, thermal effects can activate heat shock proteins to induce antitumor effects in the autoimmune system and enhance anticancer immune responses *via* exposure to heat shock protein 90 (*Zunino et al., 2016*). The peritoneal cavity is continuously perfused with a heated chemotherapy solution to provide a high intraperitoneal drug concentration (*van Ruth et al., 2004*). Higher HIPEC flow rates improve peritoneal heating efficacy and lead to more rapid heating of the peritoneum and greater peritoneal/outflow temperature gradients. Shear forces generated by fluid flow during treatment can lead directly to tumor cell death (*Furman et al., 2014*). Finally, the synergistic effect of hyperthermia and chemotherapy inhibits proliferation and induces cell death *via* the apoptotic pathway (*Cesna et al., 2018*; *Tang et al., 2006*).

## Side-effects of HIPEC in GI cancers

The decision to undergo HIPEC involves careful consideration of both the potential benefits and the possible risks of therapy including side-effects that occur if HIPEC therapy is not well controlled. These side-effects involve the digestive, hematopoietic, circulatory, metabolic, and urinary systems (Table 1 and Fig. 1). The side-effects of HIPEC vary among individuals as well as the specific agents used in the adjuvant regimen and the dose and duration of treatment.

**Table 1 Typical references of side effects of HIPEC for GI cancers.**

| Reference | Country | Type of study | No. of patients and primary site | Chemotherapeutic agents | Intraabdominal temperature and duration time | Side effects | Morbidity (Percentage, %) |
|---|---|---|---|---|---|---|---|
| Lee et al. (2022) | Korea | Retro | 124 colorectal cancer patients | MMC (35 mg/m$^2$) | 90 min at 41–43 °C | Neutropenia | 62.9 |
| Lambert et al. (2009) | USA | Retro | 117 patients appendiceal cancer patients | MMC (29.1 mg/m$^2$) | 90 min at 40 °C | Neutropenia | 39 |
| Hakeam et al. (2018) | Saudi Arabia | Retro | melphalan: 46 CIS+MMC: 35 | Melphalan (60 mg/m$^2$) OR CIS (60 mg/m$^2$) + MMC (30 mg/m$^2$) | 60 min | Leukopenia and thrombocytopenia | melphalan: 25.7/60 CIS + MMC: 17.3/68.8 |
| Kemmel et al. (2015) | France | RCT | 45 PC of colorectal cancer patients | MMC (32.5 mg/m$^2$) | 90 min at 42.5 °C | Neutropenia | 40 |
| van Vugt et al. (2015) | The Netherlands | Retro | 206 peritoneal carcinomatosis of colorectal cancer. | MMC (35 mg/m$^2$) | 90 min at 41–42 °C | Sarcopenic | 43.7 |
| Mor et al. (2022) | Israel | Retro | 191 GI cancer patients | NA | NA | Gastrointestinal anastomotic leaks | 17.8 |
| Elias et al. (2001) | France | Retro | 64 Colorectal adenocarcinomas | MMC (20 mg/m$^2$) and CDDP (200 mg/m$^2$) | 60 min at 41–44 °C | Perforation | 7.8 |
| Valle et al. (2016) | Australia | Retro | 778 peritoneal surface malignancy patients | NA | NA | Enterocutaneous fistula | 5.8 |
| Sugarbaker et al. (2006) | USA | Retro | 356 appendiceal mucinous malignancy patients | MMC; 5-Fu (600 mg/m$^2$) | 90 min at 41.5 °C | Hematological; gastrointestinal | 28; 26 |
| Levine et al. (2018) | USA | RCT | 121 Appendiceal cancer patients | MMC (40 mg) oxaliplatin (200 mg/m$^2$). | 120 min at 40 °C | Hematologic toxicity | NA |
| Ye et al. (2018) | China | Retro and cohort study | 99 peritoneal carcinomatosis patients | CP (60 mg/m$^2$) and 5-Fu (700–800 mg/m$^2$) | 60–90 min at 41–45 °C | Acute kidney injury | 90.9 |
| Kapoor et al. (2019) | USA | Retro | 23 gastric or gastroesophageal adenocarcinoma patients | CP (106.6 ± 10.9 mg/% BSA) and MMC (16 ± 1.6 mg/%BSA) | 60 min at 39–42 °C | Hypocalcemia, hypophosphatemia, and hypomagnesemia | 94%, 84% and 9.7% |
| Mangan et al. (2019) | UK | Case report | A 65-year-old man with colonic tumour | Oxaliplatin, 5-Fu and MMC | NA | Encapsulating peritoneal sclerosis | NA |
| Stewart et al. (2018) | USA | Retro | 85 appendiceal or colorectal peritoneal cancer patients | MMC and oxaliplatin | 57 min | Hyperglycemia | 86 |
| DiSano et al. (2019) | USA | Retro | 115 adenocarcinomas of gastro-intestinal origins | MMC OR MMC + CP | NA | Hyperglycemia | MMC: 39 MMC + CP: 86 |

| Table 1 (continued) | | | | | | | |
|---|---|---|---|---|---|---|---|
| Reference | Country | Type of study | No. of patients and primary site | Chemotherapeutic agents | Intraabdominal temperature and duration time | Side effects | Morbidity (Percentage, %) |
| *Smibert et al. (2020)* | Australia | Retro | 100 colorectal cancer and pseudomyxoma peritonei patients | NA | NA | Infectious complication | 43 |

**Note:**
MMC, Mitomycin C; CP, cisplatin; 5-Fu, 5-fluorouracil; CIS, cisplatin plus; CDDP, Cisplatinum; RCT, randomized trial; Retro, retrospective study.

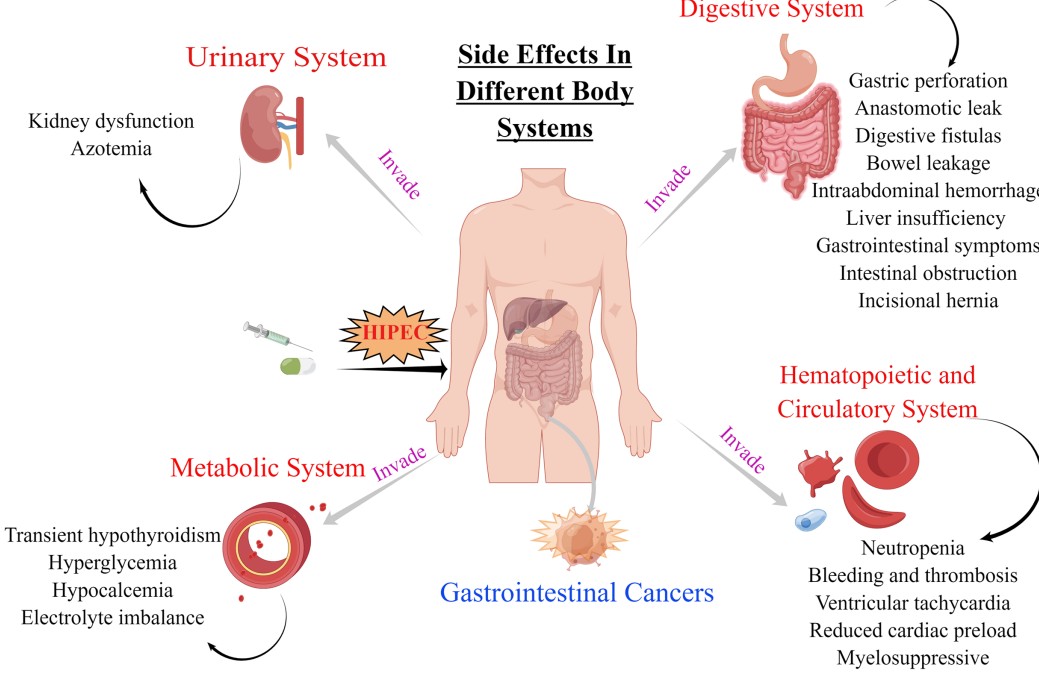

**Figure 1 Side effects of HIPEC for GI cancers in digestive, hematopoietic and circulatory, urinary and metabolic systems.** The original figure was created using figdraw (www.figdraw.com).

Substantial short- and long-term side-effects are associated with chemotherapy. Short-term side-effects include the toxic effects of chemotherapy, whereas long-term side-effects include later complications of treatment arising after the conclusion of adjuvant chemotherapy. Complications significantly affect the survival rate of patients after CRS/HIPEC. One study that summarized and analyzed the perioperative complications of 225 consecutive patients who underwent CRS/HIPEC showed that the incidence rates of low- and high-grade complications were 38.7% and 15.6%, respectively (*Tan et al., 2020*). Moreover, the survival outcomes of patients without postoperative complications were significantly better than those of patients with severe complications. In addition, intraoperative blood loss was associated with greater odds of developing postoperative complications (*Tan et al., 2020*). *Oemrawsingh et al. (2019)* found that age was also a significant risk factor, reporting a 26.7% occurrence rate of serious adverse events (grade >3) among older adults and a 10.4% rate among younger adults.

## Side-effects on the digestive system

Side-effects affecting the digestive system include gastric perforation, anastomotic leak, digestive fistulas, bowel leakage, intra-abdominal hemorrhage, liver insufficiency, gastrointestinal symptoms, and intestinal obstruction.

The effect of certain chemotherapeutic agents on wound healing and intestinal anastomosis leakage is difficult to assess in cancer patients because of the short-term survival and combination with adjuvant treatment. Previous studies identified a higher PC index, more packed cells transfused, pelvic peritonectomy, more anastomoses, and colonic resections as factors associated with GI leaks (*Mor et al., 2022*). Among the 185 patients included in that study, 16 (8.6%) developed enterocutaneous digestive fistulas, and a median of 18 days (range 9–56) was observed for spontaneous fistula closure in 14 (87.5%) patients (*Halkia et al., 2015*). *Mor et al. (2022)* reported GI leaks in 17.8% (34/191) of patients, and conservative management of GI leaks was used in most cases, whereas reoperation was required in 44.1% of the cases. *Valle et al. (2016)* reported an enterocutaneous fistula rate of 5.8% diagnosed after 13 days and a 5.7% mortality rate. Patients who had a CC2 score (nodules between 2.5 mm and 2.5 cm) cytoreduction, who had an abdominal vacuum-assisted closure device, or who smoked had a higher risk of developing a fistula.

In a study by *Zappa, Savady & Sugarbaker (2010)*, an incidence rate of 6% was reported for GI tract perforation after CRS and HIPEC. This may result from vascular compromise, delay in wound healing from chemotherapy, seromuscular tears related to traction on the stomach wall, and point pressure on the greater curvature from a long-term indwelling nasogastric tube. It may be possible to prevent this complication by reperitonealizing the greater curvature if seromuscular tears occur (*Zappa, Savady & Sugarbaker, 2010*). In HIPEC treatment, higher temperatures increased the possibility of damage to the normal tissues (*Di Miceli et al., 2012*). A prospective cohort study found that CRS-HIPEC can be associated with significantly worse bowel-related quality of life (QOL) and social function due to anxiety, embarrassment, and altered body image after the creation of a stoma (*Bayat et al., 2020*).

## Side-effects on the hematopoietic and circulatory systems

Inhibition of hematopoiesis and neutropenia are common side-effects of HIPEC, whereas thrombosis, ventricular tachycardia, and decreased cardiac preload are relatively rare. Although there are multiple reasons for the delay or dose reduction in chemotherapy, bone marrow suppression remains a major cause (*Denduluri et al., 2015*; *Soff et al., 2019*). *Lee et al. (2022)* found that MMC-induced mild and severe neutropenia occurred in 24.2% and 38.7% of the patients, respectively, and severe neutropenia developed significantly earlier than mild neutropenia and lasted significantly longer. *Schnake, Sugarbaker & Yoo (1999)* found that patients who presented with obesity and anemia had an increased risk of developing profound postoperative neutropenia, which can result in high mortality and morbidity rates. Therefore, reduced chemotherapy doses are necessary in certain patients to prevent the development of this condition (*Schnake, Sugarbaker & Yoo, 1999*). *Lambert et al. (2009)* found that the incidence of neutropenia in patients with appendiceal cancer

after MMC-HIPEC was 39%. Additionally, female sex and MMC dose per body surface area were independent risk factors for neutropenia (*Lambert et al., 2009*).

*Kemmel et al. (2015)* suggested that MMC pharmacokinetics may be a predictor of severe neutropenia in HIPEC, as plasma MMC concentrations increased 30 min (T30) and 45 min (T45) after HIPEC commencement, and neutropenia and its severity increased. *Levine et al. (2018)* found that both mitomycin and oxaliplatin were associated with minor hematologic toxicity; however, mitomycin resulted in a slightly lower QOL and higher hematologic toxicity than oxaliplatin in HIPEC. Oxaliplatin may be preferred in patients with leukopenia and mitomycin in patients with thrombocytopenia (*Levine et al., 2018*). A meta-analysis that included 3,268 patients found postoperative bleeding incidence rates within 30 days ranged from 1.7% to 8.3%, and venous thromboembolism incidence rates within 90 days ranged from 0.2% to 13.6% after CRS + HIPEC (*Lundbech et al., 2022*). *Thix et al. (2009)* reported a case of ventricular tachycardia during HIPEC with cisplatin (CP) in a patient with moderate cardiac insufficiency, which may have been caused by high plasma CP levels with concomitant low magnesium levels.

## Side-effects on the metabolic system

Metabolic system side-effects include persistent hypothyroidism, hyperglycemia, hypocalcemia, and electrolyte imbalance. *DiSano et al. (2019)* found that the rates of hyperglycemia in patients undergoing CRS and HIPEC are high, which likely represent a stress response, but do not appear to adversely affect long-term outcomes or hospital stays. *Tharmalingam et al. (2020)* reported that a patient with severe symptomatic hypocalcemia after HIPEC likely suffered from a profound inflammatory reaction with transient hypoparathyroidism, which led to symptoms of significant neuromuscular excitability. In the study by *Stewart et al. (2018)*, most patients (86%) suffered from intraoperative hyperglycemia, with values up to 651 mg/dL. Insulin was required in 66% of the patients, and 91% of the patients experienced peak hyperglycemia within an hour of perfusion, which resolved by postoperative day 1 in 91% of the patients. Hyperglycemia may is caused by using a carrier solution containing dextrose; therefore, the use of carrier solutions containing dextrose needs to be carefully considered and further investigated (*Stewart et al., 2018*).

## Side-effects on the urinary system

Renal impairment is also a common problem after HIPEC. However, whether it is caused by chemotherapeutic agent toxicity or patient kidney function changes remains controversial (*Ceresoli, Coccolini & Ansaloni, 2016*). A meta-analysis suggested that HIPEC is associated with a high risk of respiratory failure and renal dysfunction (*Desiderio et al., 2017*). In their study, *Ye et al. (2018)* found that CP application during HIPEC increased nephrotoxicity; when comparing the CP HIPEC group with the non-CP HIPEC group, urea nitrogen and creatinine levels were significantly higher in the CP HIPEC group. For patients at high risk of acute kidney injury during HIPEC treatment, strict monitoring of renal function, active diuretic therapy, and prophylactic drugs should be applied (*Ye et al., 2018*).

## Other side-effects

A number of other side-effects have been observed during or after HIPEC. Only a few cases of encapsulating peritoneal sclerosis (ERS) secondary to HIPEC have been reported. ERS is a rare surgical complication and a serious and potentially fatal complication of continuous ambulatory peritoneal dialysis that can occur after intraperitoneal treatment (*Aihara et al., 2003*; *Mangan et al., 2019*; *Takebayashi et al., 2014*). HIPEC and dimethyl sulfoxide treatment can result in scrotal ulceration with the presence of intractable and constant scrotal pain along with erythema and induration progressing to eschar (*Bartlett et al., 2019*). In a study with 206 patients, 90 (43.7%) were classified as sarcopenic. Sarcopenia was associated with a significant increase in reoperations, and skeletal muscle mass depletion was associated with an increased rate of postoperative complications in patients undergoing CRS-HIPEC for colorectal PC (*van Vugt et al., 2015*). A previous case report described a rare pulmonary complication secondary to intraperitoneal administration of MMC. Moreover, this should be considered as it was the cause of serious pulmonary toxicity. However, there was no fluid collection or other evidence of an anastomotic leak in the abdominal computed tomography scan. Therefore, an abdominal source of pulmonary toxicity was unlikely (*Abel, Kokosis & Blazer, 2017*). *Smibert et al. (2020)* reported that infections after HIPEC were noted in the surgical site, respiratory tract, and urinary tract. These included *Clostridium difficile* infection, and postoperative sepsis. In most cases, infection onset was within 7 days postoperatively, and the median length of hospitalization was 19 days (*Smibert et al., 2020*).

## Strategies for toxicity workup and management

There are several ways to address the challenges posed by the side-effects presented in this article. Recent evidence strongly suggests that an expert multidisciplinary team, including experienced surgeons and medical oncologists, should be established for HIPEC to better control hyperthermia and drug selection on an individual patient basis. One important consideration is improving the efficiency of HIPEC. A drug that enters the circulation may have little secondary therapeutic effects, but its systemic effect should be low enough to minimize its side-effects (*De Smet et al., 2013*; *Rezaeian, Sedaghatkish & Soltani, 2019*). Photothermal inorganic nanoparticles responsive to near-infrared light provide new opportunities for simultaneous and targeted delivery of heat and chemotherapeutics to tumor sites in pursuit of synergistic effects to enhance efficacy (*Zhang, Wang & Chen, 2013*).

Choosing a comparatively new and promising class of anticancer agents, such as ripretinib, for HIPEC not only improves median progression-free survival and acceptable safety profiles, but also reduces the associated adverse reactions (*Blay et al., 2020*). Neutropenia can be effectively treated with filgrastim, which is the original recombinant human granulocyte colony-stimulating factor widely used for preventing neutropenia-related infections and mobilizing hematopoietic stem cells (*Dale et al., 2018*). Assessment of platelet and leukocyte counts prior to CRS/HIPEC may help predict the development of thrombocytopenia and leukopenia (*Hakeam et al., 2018*). *Bouhadjari et al. (2016)* found that amifostine may reduce severe renal impairment when cisplatin (CP) is used in HIPEC.

*Aihara et al. (2003)* suggested that bowel obstruction does not improve with conservative treatment, and PC recurrence has been excluded through thorough examination. Moreover, an early laparotomy can resolve symptoms of bowel obstruction and restore QOL (*Aihara et al., 2003*). In addition, *Somashekhar et al. (2020)* suggested that minimizing perioperative temperatures to <36.0 °C may decrease perioperative surgical site infections in these patients after CRS and HIPEC (*Eng et al., 2018*). A study from India found that CP was a safer drug when used alone, followed by MMC, and adriamycin combined with CP had higher morbidity and worse side-effects (*Somashekhar et al., 2020*).

Traditional Chinese medicine (TCM) focuses on overall treatment of the individual, and the focus of Chinese herbal medicine is to reduce the side-effects of treatment. The various herbs used in the TCM formula are thought to have synergistic effects or reduce side-effects, that is, the characteristic of "Jun Chen Zuo Shi" of TCM formulas (*Gao et al., 2021*). TCM, such as Ginseng, Huang-Qi, BanZhiLian, TJ-48, Huachansu injection, and Shenqi Fuzheng injection, play an important role in reducing side-effects after surgery or chemotherapy by inhibiting cancer cell proliferation, regulating immunity, and suppressing angiogenesis (*Law et al., 2012*; *Li et al., 2008*; *Qi et al., 2015*). TCM may serve as a dietary herbal supplement in the treatment of GI cancers and may decrease the side-effects of chemotherapeutic agents used in HIPEC.

Regarding the management of rarely reported side-effects, such as ERS, the patient who underwent laparotomy, total enterolysis, and peritonectomy, had a satisfactory recovery, started a normal diet within 7 days, and was discharged from the hospital within 14 days after a postoperative stay without complications (*Mangan et al., 2019*). In the case of pulmonary toxicity (*Abel, Kokosis & Blazer, 2017*), the treatment with empiric antibiotics and diuretics was not effective enough, and phenylephrine and intermittent bilateral positive airway pressure were also administered for blood pressure and respiratory support. Early identification and timely use of topical mitigating agents, such as dimethyl sulfoxide (DMSO), may prevent progression to scrotal necrosis and requires surgical debridement. More effective strategies may be geared toward prevention with thorough washout following HIPEC. Preprocedural radiologic imaging or intraoperative visualization of the patent processus vaginalis, internal inguinal canal plugs, and patient education with anticipatory guidance are suggested in the event that a reaction occurs (*Abdul Aziz, Wang & Teo, 2015*; *Bartlett et al., 2019*).

There is a potential risk that the above interventions may cause other issues. Nonetheless, these measures are very effective for toxicity inspection and management in HIPEC, and the benefits of these interventions outweigh the risks of further issues arising.

## CONCLUSIONS

HIPEC is currently used as an essential component of treatment to improve the disease-free and overall survival of patients with primary and metastatic GI cancers. High complication rates are a misperception from early CRS/HIPEC experiences and should no longer deter the referral of patients to experienced centers or impede clinical trial development. However, the treatment has led to unwanted side-effects in the digestive, hematopoietic, circulatory, metabolic, and urinary systems. These side-effects vary

depending on the specific agents used in the adjuvant regimen as well as on the dose and the duration of treatment. In addition, there is considerable variability in the side-effect profile across individuals.

While HIPEC has proven to be effective in optimizing the efficacies of GI cancer treatments, traditional chemotherapy is subject to side-effects, and heat delivery is often challenging. Future studies will require tailored patient selection, timing, and optimal HIPEC regimens to improve the effectiveness of this specialized treatment for patients with GI cancer. Careful preoperative assessment of patients is paramount to ensure favorable patient outcomes following this complex procedure. Our study may also provide a rationale for concurrent treatment with drugs that protect against or compensate for the side-effects of chemotherapy. The decision to undergo HIPEC involves careful consideration of the potential benefits and risks of therapy. In future research, additional experimental and molecular epidemiological studies should explore ways to reduce the side-effects of HIPEC in patients with GI cancer.

### Funding
This work was supported by the China Postdoctoral Science Foundation (No.2022M713535). The funders had no role in study design, data collection and analysis, decision to publish, or preparation of the manuscript.

### Grant Disclosures
The following grant information was disclosed by the authors:
China Postdoctoral Science Foundation: 2022M713535.

### Competing Interests
The authors declare that they have no competing interests.

### Author Contributions
- Jiyun Hu conceived and designed the experiments, performed the experiments, analyzed the data, prepared figures and/or tables, authored or reviewed drafts of the article, and approved the final draft.
- Zhenxing Wang performed the experiments, analyzed the data, prepared figures and/or tables, and approved the final draft.
- Xinrun Wang performed the experiments, authored or reviewed drafts of the article, and approved the final draft.
- Shucai Xie conceived and designed the experiments, analyzed the data, prepared figures and/or tables, and approved the final draft.

### Data Availability
This article is a literature review and does not have raw data.

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

# PeerJ

**Desiderio J, Chao J, Melstrom L, Warner S, Tozzi F, Fong Y, Parisi A, Woo Y. 2017.** The 30-year experience—a meta-analysis of randomised and high-quality non-randomised studies of hyperthermic intraperitoneal chemotherapy in the treatment of gastric cancer. *European Journal of Cancer* **79(3)**:1–14 DOI 10.1016/j.ejca.2017.03.030.

**Di Miceli D, Alfieri S, Caprino P, Menghi R, Quero G, Cina C, Pericoli Ridolfini M, Doglietto GB. 2012.** Complications related to hyperthermia during hypertermic intraoperative intraperitoneal chemiotherapy (HIPEC) treatment. Do they exist? *European Review for Medical and Pharmacological Sciences* **16(6)**:737–742.

**DiSano JA, Wischhusen J, Schaefer EW, Dessureault S, Wong J, Soybel DI, Pameijer CR. 2019.** Postoperative hyperglycemia in patients undergoing cytoreductive surgery and HIPEC: a cohort study. *International Journal of Surgery* **64(5)**:5–9 DOI 10.1016/j.ijsu.2019.02.005.

**Dong D, Tang L, Li ZY, Fang MJ, Gao JB, Shan XH, Ying XJ, Sun YS, Fu J, Wang XX, Li LM, Li ZH, Zhang DF, Zhang Y, Li ZM, Shan F, Bu ZD, Tian J, Ji JF. 2019.** Development and validation of an individualized nomogram to identify occult peritoneal metastasis in patients with advanced gastric cancer. *Annals of Oncology* **30(3)**:431–438 DOI 10.1093/annonc/mdz001.

**Elias D, Blot F, Otmany AEl, Antoun S, Lasser P, Boige V, Rougier P, Ducreux M. 2001.** Curative treatment of peritoneal carcinomatosis arising from colorectal cancer by complete resection and intraperitoneal chemotherapy. *Cancer* **92(1)**:71–76 DOI 10.1002/1097-0142(20010701)92:1<71::aid-cncr1293>3.0.co;2-9.

**Eng OS, Raoof M, O'Leary MP, Lew MW, Wakabayashi MT, Paz IB, Melstrom LG, Lee B. 2018.** Hypothermia is associated with surgical site infection in cytoreductive surgery with hyperthermic intra-peritoneal chemotherapy. *Surgical Infections* **19(6)**:618–621 DOI 10.1089/sur.2018.063.

**Flessner MF. 2005.** The transport barrier in intraperitoneal therapy. *American Journal of Physiology-Renal Physiology* **288(3)**:F433–F442 DOI 10.1152/ajprenal.00313.2004.

**Furman MJ, Picotte RJ, Wante MJ, Rajeshkumar BR, Whalen GF, Lambert LA. 2014.** Higher flow rates improve heating during hyperthermic intraperitoneal chemoperfusion. *Journal of Surgical Oncology* **110(8)**:970–975 DOI 10.1002/jso.23776.

**Gao X, Liu Y, An Z, Ni J. 2021.** Active components and pharmacological effects of cornus officinalis: literature review. *Frontiers in Pharmacology* **12**:633447 DOI 10.3389/fphar.2021.633447.

**Garofalo A, Valle M, Garcia J, Sugarbaker PH. 2006.** Laparoscopic intraperitoneal hyperthermic chemotherapy for palliation of debilitating malignant ascites. *European Journal of Surgical Oncology* **32(6)**:682–685 DOI 10.1016/j.ejso.2006.03.014.

**González-Moreno S. 2006.** Peritoneal surface oncology: a progress report. *European Journal of Surgical Oncology* **32(6)**:593–596 DOI 10.1016/j.ejso.2006.03.001.

**Goéré D, Malka D, Tzanis D, Gava V, Boige V, Eveno C, Maggiori L, Dumont F, Ducreux M, Elias D. 2013.** Is there a possibility of a cure in patients with colorectal peritoneal carcinomatosis amenable to complete cytoreductive surgery and intraperitoneal chemotherapy? *Annals of Surgery* **257(6)**:1065–1071 DOI 10.1097/SLA.0b013e31827e9289.

**Hakeam HA, Arab A, Azzam A, Alyahya Z, Eldali AM, Amin T. 2018.** Incidence of leukopenia and thrombocytopenia with cisplatin plus mitomycin-c versus melphalan in patients undergoing cytoreductive surgery (CRS) and hyperthermic intraperitoneal chemotherapy (HIPEC). *Cancer Chemotherapy and Pharmacology* **81(4)**:697–704 DOI 10.1007/s00280-018-3537-4.

**Halkia E, Efstathiou E, Rogdakis A, Christakis C, Spiliotis J. 2015.** Digestive fistulas after cytoreductive surgery & HIPEC in peritoneal carcinomatosis. *Journal of BUON* **20(Suppl 1)**:S60–S63.

**Hamazoe R, Maeta M, Kaibara N. 1994.** Intraperitoneal thermochemotherapy for prevention of peritoneal recurrence of gastric cancer. Final results of a randomized controlled study. *Cancer* **73**:2048–2052 DOI 10.1002/1097-0142(19940415)73:8<2048::aid-cncr2820730806>3.0.co;2-q.

**Hirose K, Katayama K, Iida A, Yamaguchi A, Nakagawara G, Umeda S, Kusaka Y. 1999.** Efficacy of continuous hyperthermic peritoneal perfusion for the prophylaxis and treatment of peritoneal metastasis of advanced gastric cancer: evaluation by multivariate regression analysis. *Oncology* **57(2)**:106–114 DOI 10.1159/000012016.

**Kaibara N, Hamazoe R, Iitsuka Y, Maeta M, Koga S. 1989.** Hyperthermic peritoneal perfusion combined with anticancer chemotherapy as prophylactic treatment of peritoneal recurrence of gastric cancer. *Hepatogastroenterology* **36**:75–78.

**Kapoor R, Robinson KA, Cata JP, Owusu-Agyemang P, Soliz JM, Hernandez M, Mansfield P, Badgwell B. 2019.** Assessment of nephrotoxicity associated with combined cisplatin and mitomycin C usage in laparoscopic hyperthermic intraperitoneal chemotherapy. *International Journal of Hyperthermia* **36(1)**:493–498 DOI 10.1080/02656736.2019.1597175.

**Kemmel V, Mercoli HA, Meyer N, Brumaru D, Romain B, Lessinger JM, Brigand C. 2015.** Mitomycin C pharmacokinetics as predictor of severe neutropenia in hyperthermic intraperitoneal therapy. *Annals of Surgical Oncology* **22(Suppl 3)**:S873–S879 DOI 10.1245/s10434-015-4679-9.

**Klempner SJ, Ryan DP. 2021.** HIPEC for colorectal peritoneal metastases. *The Lancet Oncology* **22(2)**:162–164 DOI 10.1016/S1470-2045(20)30693-8.

**Kusamura S, Dominique E, Baratti D, Younan R, Deraco M. 2008.** Drugs, carrier solutions and temperature in hyperthermic intraperitoneal chemotherapy. *Journal of Surgical Oncology* **98(4)**:247–252 DOI 10.1002/jso.21051.

**LaCourse KD, Johnston CD, Bullman S. 2021.** The relationship between gastrointestinal cancers and the microbiota. *The Lancet Gastroenterology & Hepatology* **6(6)**:498–509 DOI 10.1016/S2468-1253(20)30362-9.

**Lambert LA, Armstrong TS, Lee JJ, Liu S, Katz MH, Eng C, Wolff RA, Tortorice ML, Tansey P, Gonzalez-Moreno S, Lambert DH, Mansfield PF. 2009.** Incidence, risk factors, and impact of severe neutropenia after hyperthermic intraperitoneal mitomycin C. *Annals of Surgical Oncology* **16(8)**:2181–2187 DOI 10.1245/s10434-009-0523-4.

**Law PC, Auyeung KK, Chan LY, Ko JK. 2012.** Astragalus saponins downregulate vascular endothelial growth factor under cobalt chloride-stimulated hypoxia in colon cancer cells. *BMC Complementary and Alternative Medicine* **12(1)**:160 DOI 10.1186/1472-6882-12-160.

**Lee SJ, Jeon Y, Lee HW, Kang J, Baik SH, Park EJ. 2022.** Impact of mitomycin-C-induced neutropenia after hyperthermic intraperitoneal chemotherapy with cytoreductive surgery in colorectal cancer patients with peritoneal carcinomatosis. *Annals of Surgical Oncology* **29(3)**:2077–2086 DOI 10.1245/s10434-021-10924-z.

**Levine EA, Votanopoulos KI, Shen P, Russell G, Fenstermaker J, Mansfield P, Bartlett D, Stewart JH. 2018.** A Multicenter randomized trial to evaluate hematologic toxicities after hyperthermic intraperitoneal chemotherapy with oxaliplatin or mitomycin in patients with appendiceal tumors. *Journal of the American College of Surgeons* **226(4)**:434–443 DOI 10.1016/j.jamcollsurg.2017.12.027.

**Li J, Bao Y, Lam W, Li W, Lu F, Zhu X, Liu J, Wang H. 2008.** Immunoregulatory and anti-tumor effects of *Polysaccharopeptide* and *Astragalus polysaccharides* on tumor-bearing mice. *Immunopharmacology and Immunotoxicology* **30(4)**:771–782 DOI 10.1080/08923970802279183.

**Loggie BW, Thomas P. 2015.** Gastrointestinal cancers with peritoneal carcinomatosis: surgery and hyperthermic intraperitoneal chemotherapy. *Oncology* **29**:515–521.

**Lotti M, Capponi MG, Piazzalunga D, Poiasina E, Pisano M, Manfredi R, Ansaloni L. 2016.** Laparoscopic HIPEC: a bridge between open and closed-techniques. *Journal of Minimal Access Surgery* **12(1)**:86–89 DOI 10.4103/0972-9941.158965.

**Lundbech M, Krag AE, Iversen LH, Hvas AM. 2022.** Postoperative bleeding and venous thromboembolism in colorectal cancer patients undergoing cytoreductive surgery with hyperthermic intraperitoneal chemotherapy: a systematic review and meta-analysis. *International Journal of Colorectal Disease* **37(1)**:17–33 DOI 10.1007/s00384-021-04021-6.

**Man SM. 2018.** Inflammasomes in the gastrointestinal tract: infection, cancer and gut microbiota homeostasis. *Nature Reviews Gastroenterology & Hepatology* **15(12)**:721–737 DOI 10.1038/s41575-018-0054-1.

**Mancebo-González A, Díaz-Carrasco MS, Cascales-Campos P, de la Rubia A, Gil Martínez J. 2012.** Cytoreductive surgery and hyperthermic intraperitoneal chemotherapy associated toxity in treatment of peritoneal carcinomatosis. *Farmacia Hospitalaria* **36**:60–67 DOI 10.1016/j.farma.2011.01.001.

**Mangan C, Moinuddin Z, Summers A, de Reuver P, van Dellen D, Augustine T. 2019.** Encapsulating peritoneal sclerosis following hyperthermic intraperitoneal chemotherapy. *ANZ Journal of Surgery* **89(10)**:E468–E469 DOI 10.1111/ans.14770.

**Mor E, Assaf D, Laks S, Benvenisti H, Ben-Yaacov A, Zohar N, Schtrechman G, Hazzan D, Shacham-Shmueli E, Perelson D, Adileh M, Nissan A. 2022.** The impact of gastrointestinal anastomotic leaks on survival of patients undergoing cytoreductive surgery and heated intraperitoneal chemotherapy. *The American Journal of Surgery* **223(2)**:331–338 DOI 10.1016/j.amjsurg.2021.03.061.

**Murphy N, Jenab M, Gunter MJ. 2018.** Adiposity and gastrointestinal cancers: epidemiology, mechanisms and future directions. *Nature Reviews Gastroenterology & Hepatology* **15(11)**:659–670 DOI 10.1038/s41575-018-0038-1.

**Oemrawsingh A, de Boer NL, Brandt-Kerkhof ARM, Verhoef C, Burger JWA, Madsen EVE. 2019.** Short-term complications in elderly patients undergoing CRS and HIPEC: a single center's initial experience. *European Journal of Surgical Oncology* **45(3)**:383–388 DOI 10.1016/j.ejso.2018.10.545.

**Qi F, Zhao L, Zhou A, Zhang B, Li A, Wang Z, Han J. 2015.** The advantages of using traditional Chinese medicine as an adjunctive therapy in the whole course of cancer treatment instead of only terminal stage of cancer. *BioScience Trends* **9(1)**:16–34 DOI 10.5582/bst.2015.01019.

**Rezaeian M, Sedaghatkish A, Soltani M. 2019.** Numerical modeling of high-intensity focused ultrasound-mediated intraperitoneal delivery of thermosensitive liposomal doxorubicin for cancer chemotherapy. *Drug Delivery* **26(1)**:898–917 DOI 10.1080/10717544.2019.1660435.

**Schnake KJ, Sugarbaker PH, Yoo D. 1999.** Neutropenia following perioperative intraperitoneal chemotherapy. *Tumori Journal* **85(1)**:41–46 DOI 10.1177/030089169908500109.

**Sender R, Fuchs S, Milo R. 2016.** Revised estimates for the number of human and bacteria cells in the body. *PLOS Biology* **14(8)**:e1002533 DOI 10.1371/journal.pbio.1002533.

**Seshadri RA, Glehen O. 2016.** Cytoreductive surgery and hyperthermic intraperitoneal chemotherapy in gastric cancer. *World Journal of Gastroenterology* **22(3)**:1114–1130 DOI 10.3748/wjg.v22.i3.1114.

**Smibert OC, Slavin MA, Teh B, Heriot AG, Penno J, Ismail H, Thursky KA, Worth LJ. 2020.** Epidemiology and risks for infection following cytoreductive surgery and hyperthermic intra-peritoneal chemotherapy. *Supportive Care in Cancer* **28(6)**:2745–2752 DOI 10.1007/s00520-019-05093-5.

**Soff GA, Miao Y, Bendheim G, Batista J, Mones JV, Parameswaran R, Wilkins CR, Devlin SM, Abou-Alfa GK, Cercek A, Kemeny NE, Sarasohn DM, Mantha S. 2019.** Romiplostim treatment of chemotherapy-induced thrombocytopenia. *Journal of Clinical Oncology* **37(31)**:2892–2898 DOI 10.1200/JCO.18.01931.

**Somashekhar SP, Yethadka R, Kumar CR, Ashwin KR, Zaveri S, Rauthan A. 2020.** Toxicity profile of chemotherapy agents used in cytoreductive surgery and hyperthermic intraperitoneal chemotherapy for peritoneal surface malignances. *European Journal of Surgical Oncology* **46(4)**:577–581 DOI 10.1016/j.ejso.2019.10.032.

**Spratt JS, Adcock RA, Muskovin M, Sherrill W, McKeown J. 1980.** Clinical delivery system for intraperitoneal hyperthermic chemotherapy. *Cancer Research* **40(2)**:256–260.

**Stewart CL, Gleisner A, Halpern A, Ibrahim-Zada I, Luna RA, Pearlman N, Gajdos C, Edil B, McCarter M. 2018.** Implications of hyperthermic intraperitoneal chemotherapy perfusion-related hyperglycemia. *Annals of Surgical Oncology* **25(3)**:655–659 DOI 10.1245/s10434-017-6284-6.

**Sticca RP, Dach BW. 2003.** Rationale for hyperthermia with intraoperative intraperitoneal chemotherapy agents. *Surgical Oncology Clinics of North America* **12(3)**:689–701 DOI 10.1016/S1055-3207(03)00029-2.

**Stiles ZE, Murphy AJ, Anghelescu DL, Brown CL, Davidoff AM, Dickson PV, Glazer ES, Bishop MW, Furman WL, Pappo AS, Lucas JT Jr, Deneve JL. 2020.** Desmoplastic small round cell tumor: long-term complications after cytoreduction and hyperthermic intraperitoneal chemotherapy. *Annals of Surgical Oncology* **27(1)**:171–178 DOI 10.1245/s10434-019-07339-2.

**Sugarbaker PH, Alderman R, Edwards G, Marquardt CE, Gushchin V, Esquivel J, Chang D. 2006.** Prospective morbidity and mortality assessment of cytoreductive surgery plus perioperative intraperitoneal chemotherapy to treat peritoneal dissemination of appendiceal mucinous malignancy. *Annals of Surgical Oncology* **13(5)**:635–644 DOI 10.1245/ASO.2006.03.079.

**Sugarbaker PH, Van der Speeten K, Stuart OA. 2010.** Pharmacologic rationale for treatments of peritoneal surface malignancy from colorectal cancer. *World Journal of Gastrointestinal Oncology* **2(1)**:19–30 DOI 10.4251/wjgo.v2.i1.19.

**Sung H, Ferlay J, Siegel RL, Laversanne M, Soerjomataram I, Jemal A, Bray F. 2021.** Global cancer statistics 2020: GLOBOCAN estimates of incidence and mortality worldwide for 36 cancers in 185 countries. *CA: A Cancer Journal for Clinicians* **71(3)**:209–249 DOI 10.3322/caac.21660.

**Takebayashi K, Sonoda H, Shimizu T, Ohta H, Ishida M, Mekata E, Endo Y, Tani T, Tani M. 2014.** Successful surgical approach for a patient with encapsulating peritoneal sclerosis after hyperthermic intraperitoneal chemotherapy: a case report and literature review. *BMC Surgery* **14(1)**:57 DOI 10.1186/1471-2482-14-57.

**Tan JW, Tan GHC, Ng WY, Ong CJ, Chia CS, Soo KC, Teo MCC. 2020.** High-grade complication is associated with poor overall survival after cytoreductive surgery and hyperthermic intraperitoneal chemotherapy. *International Journal of Clinical Oncology* **25(5)**:984–994 DOI 10.1007/s10147-019-01609-5.

**Tang R, Zhu ZG, Qu Y, Li JF, Ji YB, Cai Q, Liu BY, Yan M, Yin HR, Lin YZ. 2006.** The impact of hyperthermic chemotherapy on human gastric cancer cell lines: preliminary results. *Oncology Reports* **16**:631–641 DOI 10.3892/or.16.3.631.

**Tharmalingam S, Reddy S, Sharda P, Koch CA. 2020.** Severe hypocalcemia and transient hypoparathyroidism after hyperthermic intraperitoneal chemotherapy. *Hormone and Metabolic Research* **52(09)**:689–690 DOI 10.1055/a-1220-6971.

**Thix CA, Königsrainer I, Kind R, Wied P, Schroeder TH. 2009.** Ventricular tachycardia during hyperthermic intraperitoneal chemotherapy. *Anaesthesia* **64(10)**:1134–1136 DOI 10.1111/j.1365-2044.2009.05993.x.

**Valle SJ, Alzahrani N, Alzahrani S, Traiki TB, Liauw W, Morris DL. 2016.** Enterocutaneous fistula in patients with peritoneal malignancy following cytoreductive surgery and hyperthermic intraperitoneal chemotherapy: incidence, management and outcomes. *Surgical Oncology* **25(3)**:315–320 DOI 10.1016/j.suronc.2016.05.025.

**van Ruth S, Mathôt RA, Sparidans RW, Beijnen JH, Verwaal VJ, Zoetmulder FA. 2004.** Population pharmacokinetics and pharmacodynamics of mitomycin during intraoperative hyperthermic intraperitoneal chemotherapy. *Clinical Pharmacokinetics* **43(2)**:131–143 DOI 10.2165/00003088-200443020-00005.

**van Vugt JL, Braam HJ, van Oudheusden TR, Vestering A, Bollen TL, Wiezer MJ, de Hingh IH, van Ramshorst B, Boerma D. 2015.** Skeletal muscle depletion is associated with severe postoperative complications in patients undergoing cytoreductive surgery with hyperthermic intraperitoneal chemotherapy for peritoneal carcinomatosis of colorectal cancer. *Annals of Surgical Oncology* **22(11)**:3625–3631 DOI 10.1245/s10434-015-4429-z.

**Verwaal VJ, Bruin S, Boot H, van Slooten G, van Tinteren H. 2008.** 8-year follow-up of randomized trial: cytoreduction and hyperthermic intraperitoneal chemotherapy versus systemic chemotherapy in patients with peritoneal carcinomatosis of colorectal cancer. *Annals of Surgical Oncology* **15(9)**:2426–2432 DOI 10.1245/s10434-008-9966-2.

**Verwaal VJ, van Ruth S, de Bree E, van Sloothen GW, van Tinteren H, Boot H, Zoetmulder FA. 2003.** Randomized trial of cytoreduction and hyperthermic intraperitoneal chemotherapy versus systemic chemotherapy and palliative surgery in patients with peritoneal carcinomatosis of colorectal cancer. *Journal of Clinical Oncology* **21(20)**:3737–3743 DOI 10.1200/JCO.2003.04.187.

**Ye J, Chen L, Zuo J, Peng J, Chen C, Cai S, Song W, He Y, Yuan Y. 2020.** A precise temperature control during hyperthermic intraperitoneal chemotherapy promises an early return of bowel function. *Cancer Biology & Therapy* **21(8)**:726–732 DOI 10.1080/15384047.2020.1775444.

**Ye J, Ren Y, Wei Z, Peng J, Chen C, Song W, Tan M, He Y, Yuan Y. 2018.** Nephrotoxicity and long-term survival investigations for patients with peritoneal carcinomatosis using hyperthermic intraperitoneal chemotherapy with cisplatin: a retrospective cohort study. *Surgical Oncology* **27(3)**:456–461 DOI 10.1016/j.suronc.2018.05.025.

**Zappa L, Savady R, Sugarbaker PH. 2010.** Gastric perforation following cytoreductive surgery with perioperative intraperitoneal chemotherapy. *Journal of Surgical Oncology* **101(7)**:634–636 DOI 10.1002/jso.21546.

**Zhang Z, Wang J, Chen C. 2013.** Near-infrared light-mediated nanoplatforms for cancer thermo-chemotherapy and optical imaging. *Advanced Materials* **25(28)**:3869–3880 DOI 10.1002/adma.201301890.

**Zhang B, Wang H, Shen S, She X, Shi W, Chen J, Zhang Q, Hu Y, Pang Z, Jiang X. 2016.** Fibrin-targeting peptide CREKA-conjugated multi-walled carbon nanotubes for self-amplified photothermal therapy of tumor. *Biomaterials* **79**:46–55 DOI 10.1016/j.biomaterials.2015.11.061.

**Zunino B, Rubio-Patiño C, Villa E, Meynet O, Proics E, Cornille A, Pommier S, Mondragón L, Chiche J, Bereder JM, Carles M, Ricci JE. 2016.** Hyperthermic intraperitoneal chemotherapy leads to an anticancer immune response via exposure of cell surface heat shock protein 90. *Oncogene* **35(2)**:261–268 DOI 10.1038/onc.2015.82.