# Peer review of "Side-effects of hyperthermic intraperitoneal chemotherapy in patients with gastrointestinal cancers"

_PeerJ, doi:10.7717/peerj.15277_

## Round 0.1 · original submission · Major Revisions

Thank you for your interesting study, Please amend the manuscript based on the comments of reviewers.

Reviewer 1 ·

Basic reporting

The paper represents a systematic review of published data from several studies exploring the side effects of HIPEC (Hyperthermic intraperitoneal chemotherapy) for gastrointestinal cancers and proposes practical strategies for adverse event management. And classification side effects according to different systems of the human body including digestive, hematopoietic, circulatory, metabolic, and urinary systems.

Experimental design

no comment

Validity of the findings

no comment

Additional comments

1.The paper discuss some very rare side effects of HIPEC such as encapsulating peritoneal sclerosis (ERS) and scrotal ulceration, which is a good innovation.
2.This paper also provide a rationale for concurrent treatment with drugs that protect against or compensate for adverse effects in side effects resulting from chemotherapy.
3.Some suggestions were proposed to improve the quality of this manuscript. First, the paper should show more strategies for toxicity workup and management in rare side effects, and the research on traditional Chinese medicine (TCM) has attracted more and more attention around the world. So, the analysis of the effects of TCM components also provides a theoretical basis for studies of the mechanisms of Chinese herbal compounds, which contain these effective components. Therefore, your discussion needs more detail in this part. Second, please read the full text again and correct spelling and grammatical errors.
4.There are some minor comments as unsuitable spelling/format in the word, such as improper italic at line 220; and an unhandled modification symbol at line 114.

Reviewer 2 ·

Basic reporting

The focus of this manuscript is to comprehensively summarize the side effects of Hyperthermic intraperitoneal chemotherapy (HIPEC) for gastrointestinal cancers (GI cancers). HIPEC had been shown to improve the overall survival of patients with primary and metastatic GI cancers while the side effects associated hinder its application. The sides effects of HIPEC for GI cancers have not been systematically reviewed and the current study greatly advances the knowledge in the field. The manuscript is overall well-written. I only have a few suggestions for improvement.

Experimental design

The authors used Colon Cancer or Rectal Cancer as Mesh headings or key words, while many research papers used Colorectal Cancer. Could this lead to some of the studies used Colorectal Cancer not being covered by these searching criteria?

Validity of the findings

Was cytoreductive surgery considered as part of HIPEC in this study? If so, the authors should mention this and edit some of the sentences to avoid any confusion (e.g., line 59: HIPEC along with cytoreductive surgery (CRS) is the only therapeutic modality that has resulted in long-term survival in select groups of patients, line 63: HIPEC plus CRS achieved not only great survival benefits in patients with peritoneal cancer (PC) of colorectal origin). If not, were the side effects mentioned in this manuscript solely caused by HIPEC or some of them could be caused by cytoreductive surgery? In this case, “the side effects of Cytoreductive Surgery and HIPEC” should be used throughout the manuscript.

Additional comments

The authors discussed several drugs for protecting against or compensating for the adverse side effects of HIPEC. Will these drugs cause other side effects? For instance, the authors mentioned that some of the traditional Chinese medicine could be used to reduce side effects after surgery, were there any side effects caused by these medicines reported?

---

## Round 0.2 · Minor Revisions

Based on the decisions from our reviewers, the updated version of this manuscript is almost ready to be accepted for publication on PeerJ.

However, the English language must be improved before acceptance so the manuscript is being returned to the authors for a round of editing.

Reviewer 1 ·

Basic reporting

The authors have revised it according to the previous comments.

Experimental design

The manuscript is clearly written in professional, unambiguous language.

Validity of the findings

This paper describes the side effects of HIPEC for gastrointestinal cancers and proposes practical strategies for adverse event management.

Additional comments

After being revised by the author according to the previous comments, the manuscript is clearly written in professional, unambiguous language. Therefore, I recommend accepting this manuscript

Reviewer 2 ·

Basic reporting

see additional comments

Experimental design

see additional comments

Validity of the findings

see additional comments

Additional comments

The authors had revised the manuscript to address the concerns raised. They updated Mesh headings or key words to make sure studies used "Colorectal Cancer" were covered by the searching criteria. They had also revised the text to make the manuscript easier to understand.

---

## Round 0.3 · Minor Revisions

Based on the decisions from our reviewers, the updated version of this manuscript is almost ready to be accepted for publication on PeerJ.

However, the English language must be improved before acceptance so the manuscript is being returned to the authors for a round of editing.

---

## Round 0.4 · accepted · Accept

I am glad to accept this manuscript on behalf of PeerJ.